# Conflation between knowledge and acceptance may contribute to the knowledge gap between Judeo-Christian and non-religious people

Jonathan D. Hodson[1ⓔ], Dalton Bourne[1ⓔ], Noah Emery[1ⓔ], Jade B. Sorensen[1], Andrea Phillips[ⓘ2], Jamie L. Jensen[ⓘ1*]

**1** Biology Department, Brigham Young University, Provo, Utah, United States of America, **2** Department of Biology, Utah Valley University, Orem, Utah, United States of America

ⓔ These authors contributed equally to this work.
* jamie.jensen@byu.edu

## Abstract

This paper presents findings from two related studies investigating the relationship between evolution knowledge and various influencing factors, notably religious affiliation, religiosity, and evolution acceptance. Utilizing a nationwide survey study and a focused classroom study, we explored the differences in evolution knowledge agreement among individuals with different religious identifications. The nationwide study sampled 827 respondents from the United States, comprising those who identified with a Judeo-Christian religion and those who identified as agnostic/atheistic. Agnostic/atheistic respondents demonstrated notably higher evolution knowledge agreement compared to their Judeo-Christian counterparts. Structural equation modeling confirmed that evolution knowledge agreement significantly predicted acceptance within both groups. Aiming to further investigate this phenomenon, the classroom study involved non-major introductory biology students in a religiously homogeneous institution. Altering survey question wording to mitigate potential conflict revealed a substantial increase in measured evolution knowledge agreement among highly religious Christian students simply by prefacing questions with "according to science." This finding suggests that the gap between their endorsement of evolution knowledge and acceptance may be partly due to respondents intentionally or unintentionally masking their true understanding to align with their religious beliefs. Overall, this study presents evidence suggesting that evolution knowledge may be conflated with acceptance, highlighting the importance of nuanced approaches to addressing evolution acceptance within diverse communities.

## Introduction

Though evolution is one of the foundational principles of modern biology [1], it is widely rejected worldwide as the best explanation for life on earth [2–5]. The United

**Data availability statement:** Data is available through BYU ScholarsArchive at https://scholarsarchive.byu.edu/data/86.

**Funding:** Internal University funding. The funders had no role in study design, data collection and analysis, decisions to publish or preparation of the manuscript.

**Competing interests:** The authors have declared that no competing interests exist.

States has some of the lowest evolution acceptance rates, especially regarding human evolution, with 40% accepting Young Earth Creation (earth and its inhabitants created by a divine entity as described in the Bible) [5]. When given the more nuanced option of a theistic evolutionary explanation from the start ("Human beings have developed over millions of years from less advanced forms of life, but God guided this process"), this number drops to 18% [5]. This deficit in nationwide acceptance of evolution has led researchers to seek out the underlying factors. Prior research has shown that Christian religious affiliation is a predictor of evolution rejection [6]. Only 64% of Protestants, 38% of Evangelicals, and 42% of members of the Church of Jesus Christ of Latter-day Saints accept human evolution [3], compared to 98% of members of the American Association for the Advancement of Science [4]. Additional studies have shown religiosity, defined as an individual's level of commitment to and practice within their religion, to be negatively correlated with the acceptance of evolutionary theory [6–9]. In summary, according to national polls, one's affiliation with and commitment to Christian, or biblically grounded, religious belief is likely to hinder the acceptance of evolution in the United States context. For the purposes of this study, the authors define evolution acceptance as a stance accepting statements consistent with evolution as the best explanation for the presence of present biodiversity, inclusive of microevolution, macroevolution, and human evolution. We define evolution rejection as the acceptance of any other explanation for biodiversity on Earth to the exclusion of evolutionary principles, e.g., Special Creation as outlined in the Bible. These definitions are informed by prior research illustrating the selective acceptance of evolution among religious individuals, especially with regards to human evolution [10].

More recent research investigating this phenomenon suggested that the relationship between evolution and religion is more complex than previously assumed. Researchers have proposed that it is not necessarily one's involvement with a religious faith that will determine their rejection or acceptance of evolution, but rather one's perceived conflict between their religious worldview and evolutionary theory [10]. For example, highly religious individuals could readily accept evolutionary theory in the case that they do not perceive significant conflict between evolution and their religious beliefs. Indeed, research has shown that the most significant predictors of evolution rejection are perceived conflict between evolution and belief in God, and perceived conflict between evolution and personal religious beliefs, even when religiosity, religious affiliation, and evolution understanding were controlled for [10].

Other key factors besides religious affiliation and religiosity have also been shown to predict acceptance of evolution. One such factor is one's understanding of the nature of science [6,11,12]. The better an individual understands how scientific understanding is produced, the more likely they will be to accept evolution as a scientific theory [13]. Cofré et al. [14] found that teachers who participated in a workshop on the nature of science showed higher evolution acceptance as their understanding of the nature of science improved. When a similar understanding is established among students, it acts as an important foundation that increases

their acceptance of the scientific validity of evolution [15]. It is important to note that despite this link between nature of science understanding and evolution acceptance, scientific reasoning skills (i.e., one's ability to use science process skills) have been shown to be unrelated to acceptance rates [16]. Another influential factor is knowledge of evolution. Researchers who have sought to measure evolution knowledge— how well a student understands specific principles of evolution—have found a clear relationship between a student's knowledge of evolutionary principles and their acceptance of the theory [17–19]. These findings have led some researchers to propose a deficit model [20,21] suggesting that the only barrier to students' acceptance of evolution is knowledge, or a resolution model that encourages instructors to utilize a "just facts" approach to close the proposed knowledge gap [22–26]. Some have proposed, based on the link between knowledge and acceptance, that students who reject evolution are simply irrational or uneducated [27]. In a large-scale study of the U.S. population, researchers aimed to establish more concretely the relationship between evolution knowledge and acceptance with a more comprehensive knowledge measure than had previously been used [28]. With a large population, it was shown that higher knowledge of evolution was predictive of higher acceptance. Additional research, however, has rejected these findings in favor of a lack of correlation between evolution knowledge and acceptance [29–34]. Indeed, research has shown that once religious affiliation and political ideology are controlled for, education level is not a factor in evolution acceptance [2,35]. Examining the relationship between religiosity and acceptance, research parsing out evolution knowledge and religiosity into different contexts (e.g., microevolution, single universal common ancestor) showed that religiosity had a different impact depending on the specific claims being made regarding evolution; the impact of religiosity was more complex than previously determined [36]. Prior research has also shown the complexity of determining the impact of knowledge on acceptance; in populations where evolution acceptance is high, knowledge may be sufficient to further increase acceptance [37], whereas in populations where acceptance is very low, sharing facts is ineffective [38]. These contradictory and context-dependent findings point to the need for more research, especially that seeks out potential mediating factors to the knowledge and acceptance phenomenon. These studies set out to fill this gap in the research on evolution acceptance when it comes to interpreting the relationship between knowledge of evolution and acceptance of evolution. The first study presented in this manuscript aimed to establish clear relationships between evolution knowledge agreement, religiosity, and evolution acceptance. When these results revealed that knowledge is less predictive of acceptance among religious individuals, a second classroom study was employed to test further a possible explanation for the discrepancy. Given the link between religious affiliation and evolution rejection [6], and the link between religiosity and evolution rejection [9], we hypothesize that evolution knowledge may be conflated with evolution acceptance--participants' responses to knowledge instruments may be influenced by whether or not they are accepting of evolution as a result of their religious beliefs. One of the reasons for this may be the existence of conflict-induced identity-protective cognition [39]. This arises when an individual perceives cognitive conflict between two constructs (such as evolution and religious belief) and errs on the side of willingly disengaging rather than reconciling or rejecting in order to protect their previously held identity. A second reason may be that religious individuals feel stigmatized in the science sphere [40], and thus unwillingly disengage from it.

As previously stated, the purpose of these studies was to further assess the relationship between evolution knowledge and acceptance. When the first, nationwide survey study resulted in additional questions, a second, classroom-scale study was conducted to test the hypothesis that evolution knowledge and evolution acceptance are conflated. Based on theoretical rationales, we predict that conflation is arising because religious participants are either willingly disengaging (or masking their knowledge) in order to maintain their prior identity when feelings of conflict arise [39], or they are unwillingly disengaging due to feeling stigmatized in a science sphere [40] or due to the prompting of a science schema to which they have disagreement. By examining knowledge and acceptance of evolution in these two different populations, we hoped to add nuance to the general trends seen in large-scale survey data with smaller-scale classroom data targeting the conflation we hypothesized exists between these constructs.

## Study 1

The purpose of study 1 was to test our hypothesis that there is a relationship between evolution knowledge agreement and evolution acceptance, and to test the hypothesis that this relationship may be mediated by religious identity. To test these, we conducted a nationwide sampling of individuals who either identified as agnostic/atheist or who identified with a Judeo-Christian religion.

### Methods

**Ethics statement.** This study was approved by the authors' Institutional Review Board, #IRB2022−392 and #IRB2023−104. All respondents gave permission for the anonymous use of their data. Respondents were included between October 27, 2022 and January 9, 2024.

**Sample population.** This study examined the relationship between evolution knowledge and acceptance on a large scale through administration of a survey targeting evolution knowledge, evolution acceptance, and religiosity, along with several demographics. Through the Qualtrics® surveying platform [41], using their sample panels, we requested 800 responses with the following demographics: 400 respondents who identified with one of seventeen Judeo-Christian religious affiliations (including "other Christian"), and 400 respondents who identified as agnostic or atheist; over 18 years of age; and residing in the United States. Additionally, we asked for a stratified random sample representative from their "General Population" sample, with an additional filter of 50% some college or less, and 50% associates degree or above. We included several attention checks within the survey, which consisted of questions that instructed respondents which answer to select to help ensure their answers were not random. We obtained 882 responses. Fifty-five respondents were removed for failing attention checks or not providing complete responses, leaving 827 respondents for analysis. Additional demographic information was collected via survey questions including education level (high school to graduate/professional degree), gender, and political ideology. A summary of sample demographics is included in Table 1. (The full survey is accessible in S1 File).

We acknowledge that not all Judeo-Christian religious groups identify similarly in their acceptance of evolution. We analyzed acceptance and knowledge between religious groups. Where significant differences between groups existed, we eliminated groups with ten or fewer representatives and found that resulting relationships did not change. We also considered differences in our outcome measures between agnostic and atheist respondents. Where differences existed, we again found that resulting relationships did not change by disaggregation. Thus, we maintained all Judeo-Christian groups together and agnostic and atheist respondents together in our analyses. Differences are reported in Table 2. Two respondents identified as Muslim and were eliminated from analyses.

**Survey instruments.** Evolution knowledge agreement was measured using five items from the "Evolutionary Knowledge" subscale of the Evolution Attitudes and Literacy Survey-Short Form (EALS-SF; [42]). The only instructions preceding the items was the statement, "To what extent do you agree with the following statements?" Items were measured on a 6-point Likert scale from "strongly disagree" to "strongly agree" with no neutral option. Providing a neutral response may decrease validity in studies on controversial topics where social cohesion and acceptance play a role [43]. Students provided a neutral response option may opt for neutrality instead of choosing what they perceive to be a socially undesirable response. Because of the social component of faith and religious involvement, we chose to exclude the neutral response option from the Likert scale to limit potential response bias. The knowledge measure for each respondent was calculated by summing their responses from these items. We acknowledge, however, that because this instrument is presented in Likert-style, it is measuring agreement with knowledge statements (hereafter referred to as "evolution knowledge agreement") rather than actual knowledge that might be assessed by a multiple-choice format. This was critical to our hypothesis that in certain knowledge instruments, students may be conflating scientific knowledge with acceptance, hence our intervention of adding "according to science."

**Table 1. Demographic description of nationwide sample.**

| | Agnostic/Atheist Respondents (n = 421) | Judeo-Christian Respondents (n = 408) |
|---|---|---|
| Education | | |
| High school only | 9.3 | 18.4 |
| Some college | 26.6 | 25.7 |
| Associates degree | 22.9 | 10.3 |
| 4-year college degree | 34.7 | 23.8 |
| Graduate/Professional degree | 17.6 | 21.8 |
| Gender | | |
| Male | 41.6 | 37.5 |
| Female | 55.1 | 62.3 |
| Non-binary/third gender | 2.9 | 0.2 |
| Prefer not to say | 0.5 | 0 |
| Ideology | | |
| 1 (Conservative-leaning) | 1.5 | 10.4 |
| 2 | 4.9 | 21.0 |
| 3 | 6.1 | 18.2 |
| 4 (Moderate) | 19.8 | 25.5 |
| 5 | 19.2 | 10.4 |
| 6 | 28.1 | 9.2 |
| 7 (Liberal-leaning) | 20.5 | 5.4 |

Evolution acceptance was measured using the Inventory of Students' Acceptance of Evolution (I-SEA; [44]). This instrument measured acceptance of three constructs: Microevolution, Macroevolution, and Human Evolution, each with 8 items on a six-point Likert scale, again with no neutral option to avoid noncommittal responses. Again, the only instructions preceding the items was the statement, "To what extent do you agree with the following statements?" Total acceptance for each participant was calculated by summing all items after reverse coding necessary items, resulting in higher scores reflecting higher levels of acceptance.

Religiosity was measured using a modified version of a religiosity instrument by Sethi & Seligman [45] validated in previous studies [16,46]. It measured the latent variables religious hope, religious influence, and religious practice, each with five items. Political ideology was measured using one item that asked, "What is your political ideology?" on a 7-point Likert scale from "Very Conservative" (at 1) to "Very Liberal" (at 7). And education level was measured with one item, having the options "High school only", "Some college", "Associates degree", "4-year college degree", and "Graduate/Professional degree".

We administered a test of scientific reasoning to the nationwide sample, Lawson's Classroom Test of Scientific Reasoning (LCTSR; [47], ver. 2000). However, several factors (including non-discriminating items, responses that did not differ from guessing, and one typographical error) rendered the instrument invalid for analyses. A previous study on a robust sample similar to the current sample, however, showed no relationship between this instrument and evolution acceptance [16], thus we omitted the instrument. The ordering of the survey was consistent for all respondents and was as follows: demographics, scientific reasoning, religiosity, I-SEA, EALS-SF.

**Survey validation.** Latent variables were confirmed using confirmatory factor analysis (CFA) (see S1 Table for factor loadings). All latent variables were represented by at least three indicators. We performed CFA with request for modification indices and removed items that did not have acceptable fit statistics [root mean square error approximation (RMSEA), comparative fit index (CFI), Tucker-Lewis index (TLI), and standardized root mean square residual (SRMR); thresholds were set at <.08, >.90, >.90, and <.08, respectively.] An MLR estimator was used for Likert scales and a WLSMV estimator was used for the dichotomous reasoning test items.

**Table 2. Sample size, evolution knowledge, and evolution acceptance between agnostic, atheist, and different religiously affiliated respondents.**

| Affiliation | Sample Size | Knowledge *M*(*SD*) | Acceptance *M*(*SD*) |
|---|---|---|---|
| Agnostic | 242 | 20.1 (2.8) | 118.2 (16.3) |
| Atheist | 179 | 19.7 (3.8) | 121.4 (16.1) |
| Baptist | 38 | 17.3 (4.0) | 8.3 (30.5) |
| Catholic | 102 | 18.1 (3.1) | 103.9 (19.2) |
| Christian | 93 | 17.4 (3.3) | 88.5 (23.9) |
| Church of Christ/Disciples of Christ | 6 | 19.2 (2.3) | 70.0 (9.3) |
| Church of Jesus Christ of Latter-day Saints | 36 | 20.5 (2.0) | 112.2 (17.0) |
| Congregational | 1 | 18.0 (n/a) | 74.0 (n/a) |
| Episcopalian/Anglican | 7 | 18.1 (2.8) | 109.9 (17.8) |
| Jehovah's Witness | 4 | 17.8 (2.6) | 54.3 (5.6) |
| Jewish | 17 | 18.1 (3.8) | 101.4 (27.5) |
| Lutheran | 27 | 17.2 (2.5) | 100.9 (21.1) |
| Methodist/Wesleyan | 23 | 18.6 (3.2) | 107.3 (16.4) |
| Orthodox (Eastern) | 3 | 18.3 (3.8) | 108.7 (29.7) |
| Pentecostal/Charismatic | 10 | 17.1 (1.7) | 61.9 (18.3) |
| Protestant (Other) | 30 | 18.0 (2.4) | 93.4 (26.2) |
| Reformed/Presbyterian | 3 | 22.3 (2.1) | 125.7 (16.4) |
| Seventh-day Adventist | 2 | 17.0 (4.2) | 75.0 (2.8) |
| Other Christian not listed above | 3 | 16.7 (1.5) | 97.7 (13.3) |

**Population statistical comparisons.** We used independent samples t-tests to compare knowledge between agnostic/atheist individuals and Judeo-Christian individuals in the nationwide sample. We used SPSS statistics software (IBM 2021, Armonk, NY, USA) for these analyses.

**Model testing.** To test our models of relationships, we performed Structural Equation Modeling (SEM) and Analysis of Variance or Covariance (ANOVA/ANCOVA). A MLR estimator was used for SEM analyses. We used Mplus software, ver. 8 (Muthen & Muthen, 1998–2001, Los Angeles, CA, USA) for both CFA and SEM analyses.

## Results

**Knowledge of evolution.** CFA was run on the five evolution knowledge items from the EALS instrument. Fit statistics indicated that it measured a single latent variable (RMSEA = .099, CFI = .989, TLI = .977, SRMR = .015). All factor loadings were above 0.5, except item 1 with a factor loading of .360. We removed this statement. We tested for measurement invariance between survey respondents who identified as Judeo-Christian and those who identified as agnostic or atheistic. We failed to reject the null (i.e., measurement invariance held) given a configural CFI of .986, a metric CFI of .991, and a scalar CFI of .990.

Knowledge agreement among different religious groups differed slightly [$F(1,389) = 7.14$, $p < .001$; see Table 2]. However, many groups did not have a representative sample to warrant individual analyses. Knowledge agreement between agnostic and atheist respondents did not differ ($p = $ NS). Thus, we kept both groups combined. An independent samples t-test revealed that those respondents who identified as agnostic or atheist scored higher on the knowledge questions than those who identified with one of the Judeo-Christian religions combined ($M_{ag/ath} = 24.00$, $SD = 3.92$; $M_{J-C} = 21.94$, $SD = 3.69$ on a 30-point scale; $t(827) = 7.79$, $p < .001$).

**Acceptance of evolution.** A CFA on the full I-SEA showed adequate fit (RMSEA = .069, CFI = .911, TLI = .901, SRMR = .045). Fit statistics indicated that it measured three latent variables: microevolution, macroevolution, and human evolution. All factor loadings were above 0.5. We tested for measurement invariance between survey respondents who identified as religious and those that identified as agnostic or atheistic. We rejected the null hypothesis of invariance given a configural to metric variation in CFI of greater than.01 (.892 to.877, respectively; [48]) This indicated that the factor structure was different between Judeo-Christian individuals and agnostic/atheist individuals. Thus, we ran CFA analyses on the I-SEA and subsequent SEM models separately for each group. Within Judeo-Christian individuals, human evolution item 8 was removed for lack of fit; the remaining items had adequate fit (RMSEA = .071, CFI = .918, TLI = .907, SRMR = .055). Within the agnostic/atheist group, macroevolution item 8 was removed for lack of fit; the remaining items had adequate fit with all items included (RMSEA = .061, CFI = .908, TLI = .895, SRMR = .047). Therefore, total acceptance for each group was calculated by summing the remaining 23 items for a total of 138.

Again, differences were seen between different religious affiliations in acceptance (see Table 2), however, many groups did not have a representative sample to warrant individual analyses. Acceptance was slightly higher in atheist respondents ($M = 121.39$, $SD = 16.14$) than agnostic respondents ($M = 118.18$, $SD = 16.28$, $t(419) = 2.01$, $p = .045$), but they were practically identical. We chose to keep all religious groups together and agnostic and atheist respondents together for further analysis.

Agnostic/atheist respondents reported high acceptance of evolution ($M = 119.55$, $SD = 16.28$, on a 138-point scale). Judeo-Christian respondents reported a much lower acceptance of evolution ($M = 96.32$, $SD = 24.93$, on a 138-point scale). Due to the high acceptance of evolution among Agnostic/atheist individuals (i.e., they maxed out the score on the instrument), we failed to confirm measurement invariance on the acceptance instrument between agnostic/atheist respondents and Judeo-Christian respondents. Because of this, more detailed analysis would require an instrument better able to differentiate between high and very high levels of acceptance. Acceptance between groups broken into macroevolution, microevolution, and human evolution is shown in Fig 1.

**Demographics.** Religiosity. Religiosity was only taken into account for Judeo-Christian respondents. Religious hope proved to be an unreliable measure with poor fitting items and low factor loadings. Thus, it was removed from the measure. Religious practice and religious influence as measures of religiosity showed adequate fit (RMSEA = .079, CFI = .962, TLI = .918, SRMR = .065).

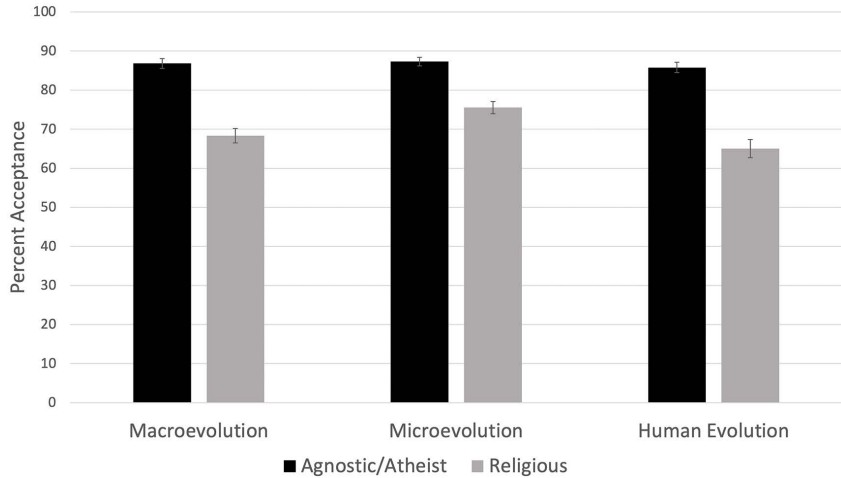

**Fig 1. Evolution acceptance compared between agnostic/atheist and religious individuals.** Error bars represent 95% confidence intervals.

Political Ideology and Education Level. An independent-samples t-test indicated that agnostic/atheistic respondents leaned farther to the left, politically, than Judeo-Christian respondents [Mag/ath = 5.17, SD = 1.48; MJ-C = 3.54, SD = 1.65 on a 7-point scale; t(809) = 14.81, p < .001]. Additionally, agnostic/atheistic respondents had slightly higher education levels than Judeo-Christian respondents [Mag/ath = 3.25, SD = 1.28; MJ-C = 3.04, SD = 1.45 on a 5-point scale; t(825) = 2.16, p = .031], a 3 being an associates degree, and a 4 being a 4-year college degree.

**Relationship between knowledge and acceptance.** Structural equation modeling showed that knowledge agreement significantly predicted acceptance within each group. Among agnostic/atheist respondents (see Fig 2), two covariates were included: political ideology and education level. While education level showed no relationship with any acceptance measures, the more liberal-leaning a respondent's ideology, the higher their acceptance of evolution in all three categories. Among Judeo-Christian respondents (see Fig 3), we included religiosity measures along with political ideology and education level. The model was run with the full array of religious affiliations and again with only those affiliations who had more than ten representatives to account for potential differences in religiosity. The relationships remained the same, so the full data set was maintained. The model showed that religious influence was a significant negative predictor of all measures of acceptance, but religious practice had no relationship. More liberal-leaning ideology predicted higher acceptance of macroevolution and human evolution, but not microevolution. Additionally, education level was shown to be a small, but significant predictor of microevolution and human evolution acceptance. Neither education level or religious influence were predictive of knowledge level among Judeo-Christian respondents.

## Discussion

We found that agnostic/atheist respondents agreed more with evolution knowledge items on average than those who identify with a Judeo-Christian religion in the United States. Given that a previous study suggests scientific reasoning ability is not predictive of religiosity (e.g., [16]), it is unlikely that lower evolution knowledge is a direct result of lower reasoning

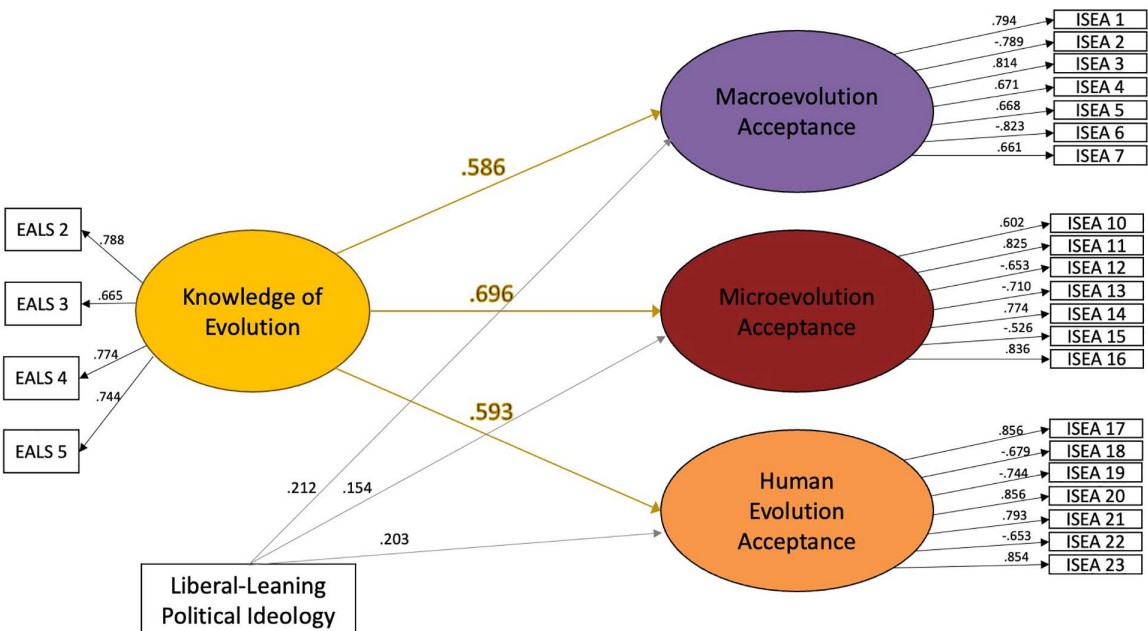

**Fig 2. Structural equation models showing the relationship between knowledge and acceptance for agnostic/atheistic respondents.** Numbers represent standardized loading factors. ** indicates significance of p < .001.

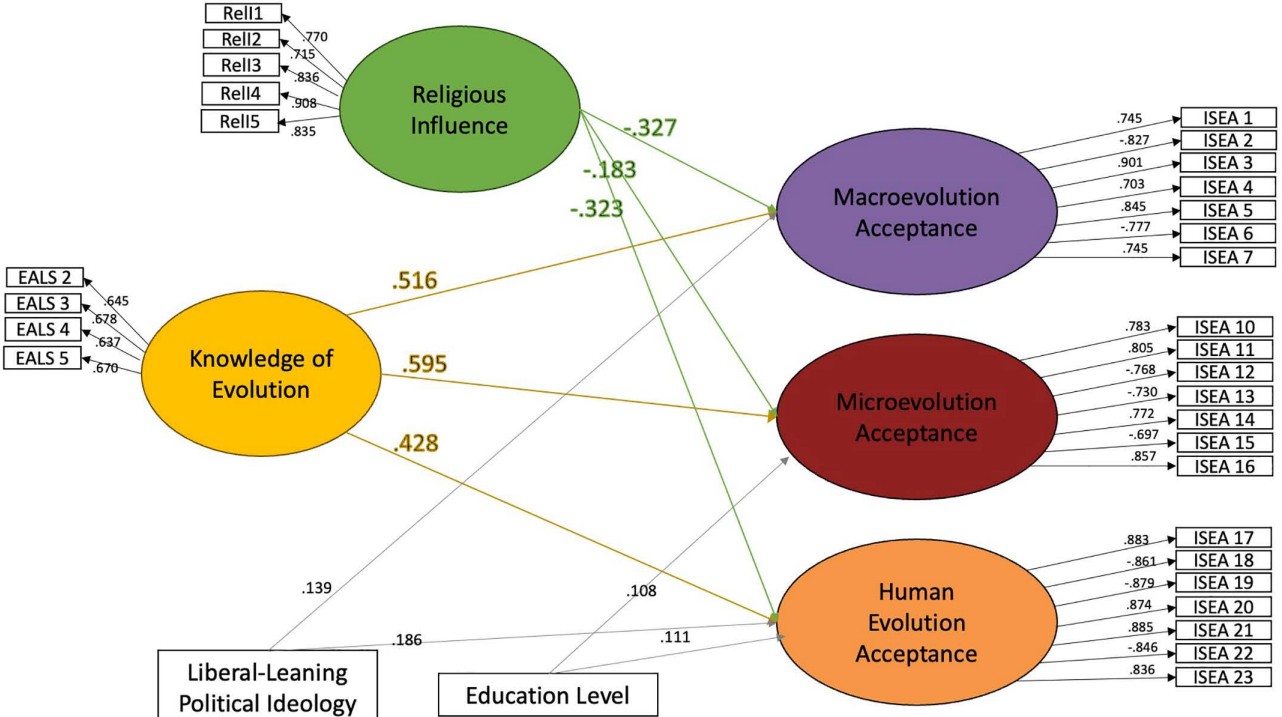

**Fig 3. Structural equation models showing the relationship between knowledge and acceptance for judeo-christian respondents.** Numbers represent standardized loading factors. ** indicates significance of $p < .001$.

ability among religious people. We therefore hypothesized, in alignment with previous studies, that other factors influence evolution knowledge other than reasoning ability. Education level was a small predictive factor in evolution acceptance among Judeo-Christian respondents. However, it was not predictive of knowledge levels indicating that something else about receiving an education may be influencing acceptance levels. Perhaps it is due to an exposure to a diversity of ideas through education that may lead to more open-mindedness to ideas like evolution, as was suggested by a recent study on university students showing increased open-mindedness with years in higher education [49]. Additionally, political ideology is likely an influential factor, with more liberal viewpoints favoring a more scientific viewpoint, as has also been previously shown [50]. The classroom study provides an opportunity to investigate possible explanations for the knowledge gap between agnostic/atheist and Judeo-Christian respondents.

## Study 2

When the first, nationwide study resulted in questions about possible conflation between evolution knowledge and acceptance, a similar survey was administered to a large group of university students with a critical wording change when asked to report on evolution knowledge: "According to science…" The purpose of adding this phrase was an attempt to mitigate feelings of perceived conflict on behalf of the respondents by discursively separating scientific understanding from personal belief.

### Methods

**Sample population.** The classroom population was recruited from four sections of a biology course (a general requirement) that took place over two semesters at a large (approx. 35,000 students), private, religious university in the Western United States. The vast majority of the university participants identified as members of the same conservative,

Christian religion. Students ranged from freshmen to seniors. Typically, the course is taken by non-biology majors (although some may be in other STEM fields) and is usually their first exposure to college-level biology. Though demographics were not gathered on this specific population, we can assume that the general biology students are representative of the university population consisting of approximately 51% female, 81% white, 9% Hispanic, 1% Pacific Islander, and 1% black individuals, with the middle 50% high school GPA being 3.86–4.00, and the middle 50% ACT score ranging from 27 to 32.

The classroom population was split into a control group and a test group that were administered modified versions of the survey. The control survey was administered in two sections (n = 384) and the test survey (which included the statement "According to science…" in front of the questions related to evolution) was administered in two sections (n = 385) via Qualtrics® prior to the start of the evolution unit (i.e., they had not been exposed to evolution in this class previously).

**Survey instruments.** All students were administered the EALS-SF, I-SEA, and Religiosity surveys as described above, with some exceptions. In the control section (N = 384), students were given the original EALS-SF questions on evolution knowledge, while in the test section (N = 385), students were given the EALS-SF questions with the phrase, "According to science…" preceding each item. All five items were retained.

The I-SEA survey in the classroom sample we a subset of the items, excluding several items that were historically poor indicators of acceptance in this population in previous studies (see [51]). Similar to evolution knowledge, total acceptance was calculated by summing all included items. The religiosity instrument was administered and tallied in identical fashion to Study 1.

**Statistical analyses.** We used independent samples t-tests to compare knowledge and acceptance between treatments in our classroom study. We used SPSS statistics software (IBM 2021, Armonk, NY, USA) for these analyses.

**Results. Group equivalence measures:** To ensure that each treatment group was as equivalent as possible, we gathered their evolution acceptance using a subset of the I-SEA shown to be effective on this population [51]. An independent samples t-test showed equivalence of acceptance [$M_{control}$ = 36.05, $M_{test}$ = 35.89, $t(760)$ =.44, $p$ = .66].

**Effect of treatment on measured knowledge:** By adding "According to science…" preceding each knowledge statement of the EALS-SF in the treatment group, evolution knowledge increased from an average sum of 23.2 (out of a possible 30) in the control to 24.2 in the treatment group [$t(767)$ = 4.74; $p < .001$; see Fig 4].

**Discussion.** The classroom study data indicated higher knowledge of evolution agreement among those who were provided with the modified survey items on evolution knowledge that contained the phrase "According to science…"

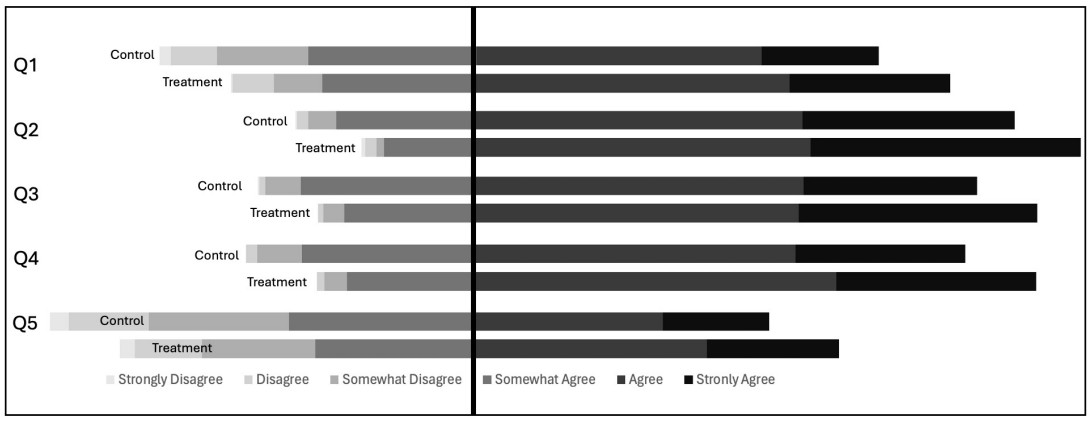

**Fig 4. Shift in Likert-scale responses to EALS items.** The darker the bars, the more respondents agreed with the statements. Students in the control group are the top bars; students in the treatment condition are the bottom bars.

This points to potential conflation between knowledge and acceptance, at least in the instruments used in this study, as the substance of the questions targeting evolution knowledge did not change, but rather the indicated source of the knowledge.

**General discussion**

In this study, we show a difference in evolution knowledge agreement between Christian and atheist/agnostic respondents in a nationwide sample. In the classroom follow-up, we found that adding "According to science…" to the knowledge questions, we could inflate agreement within religious students. In an effort to explain this phenomenon, we discuss three potential explanations. The first, as present in the existing literature, is that Christian individuals have a lack of understanding that stems from lower engagement in the process of learning due to potential conflict-induced identity-protective cognition [39], i.e., they are *willingly* disengaging to avoid feelings of conflict and protect, either consciously or subconsciously, their religious identity. In a recent study, Barnes, Supriya, and colleagues [10] found that perceived conflict was a better predictor of evolution rejection than actual religious involvement. Cognitive dissonance theory [52] posits that individuals have an inner drive to maintain cognitive consistency between their beliefs or opinions and knowledge such that if there is incongruence among them, the resulting tension drives the individual to either replace incongruence or unknowns with current and preferred understanding of the world, or avoid information that disrupts their mental schema unless it is sufficiently convincing. The consequence of this dissonance is identity-protective cognition [39]. In this case, individuals perceive the acceptance of any ideas or information contrary to the beliefs of their religious community as a failure to be faithful to their community and will therefore adopt beliefs consistent with their in-group and ignore contrary factual information, particularly if it originates from an "out-group" source. As a consequence, it is possible that religious individuals have disengaged from learning about evolution, resulting in lower levels of knowledge.

The second explanation, also present in the literature, is that religious individuals lack evolution understanding because they have been actively excluded from the learning process by stigmatization, i.e., they are *unwillingly* disengaging. Based on the theory of concealable stigmatized identities [40,53], this stigmatization occurs when people perceive themselves as devalued, or are directly devalued by others because of their particular identity [40,54,55]. Religious affiliation has been shown to be a concealable stigmatized identity in academic biology [56]. In fact, Christians have been stereotyped as having lower abilities in science, which may cause Christians to experience stereotype threat and disidentify with science [57]. This stigmatization has contributed to a lower sense of belonging [58] and higher anxiety [59], which can naturally lead to exclusion and lower achievement [60–62] among religious individuals. Thus, religious individuals may have been actively excluded by stigmatization, leading to lower levels of understanding of evolution as they unwillingly disengage from evolution learning.

The third explanation, as is illustrated by the data from the classroom data in this study, is that the gap in knowledge agreement is a manifestation of the conflation of knowledge and acceptance such that religious individuals may be purposefully or inadvertently masking their true understanding. In an anonymous survey, respondents may choose their beliefs (i.e., a rejection of evolutionary theory) over a demonstration of their knowledge, especially given that there were no consequences of disagreeing with a knowledge statement (see [6–8,28]. In the university context where the classroom study took place, graduating seniors score in the 96th percentile of all universities on the ETS Biology Field Exam on subsection 4 (Population, Evolution, & Ecology) and in the 99th percentile on the Population Genetics and Evolution Assessment Indicator (AI7; 68 compared to a national average of 41.9; unpublished institutional data), yet display rates of between 60 and 75% on evolution acceptance instruments [51]. This pattern holds in the classroom study data as well–in the control section, acceptance scores often matched knowledge scores; for example, one respondent who scored 57% on the acceptance measure scored a corresponding 53% (or raw total of 16 out of 30) on the knowledge agreement measure. However, in the test section (where "According to science" was added), there were cases of extreme disconnect, such as a respondent who scored 26% on acceptance, but 83% (or raw total of 25 out of 30) on knowledge. It is important

to note that these instruments are measured on a Likert scale, so percentages or totals are less informative of an actual cognitive level but reflect a relative acceptance or rejection. In these two cases, we see acceptance levels closely mirroring one another in the control condition, and often widely disparate in the treatment condition. These results specifically seem to point to the discrepancy between acceptance and knowledge of evolution. Using high-achieving and highly religious students, we found that evolution knowledge agreement scores significantly improved simply by adding the phrase "according to science" in front of each question. Adding the phrase "according to science" may have mitigated perceived conflict between scientific and religious understanding allowing students to separate the statement from any admission of personal belief. This would suggest that religious students, attempting to defend their religious convictions, purposely answer certain questions incorrectly even if they know the correct answer on the original instrument. Alternatively, they may be unconsciously or unintentionally revealing knowledge in response to the prompting of a different schema by the words "according to science". In other words, without this phrasing, students may have seen these knowledge questions through the same schema as the acceptance questions; but when prompted by the words "according to science", it may have activated their "science schema" allowing them to activate their actual knowledge. Additionally, it is possible that the phrase "according to science" may have invoked negative feelings by students who do not identify with science causing them simply to choose the answer opposite of what they think. Without interviewing students to understand their reasoning, it is impossible to disentangle these possibilities. However, it is clear that with the current instrumentation, students seem to be conflating their knowledge with their acceptance.

If evolution knowledge has become conflated with evolution acceptance, as our findings suggest, more research is needed to understand what factors might influence evolution acceptance among individuals across the nation. As previously discussed, evolution acceptance is the result of a complex interplay of factors. Knowledge, as one of these factors, certainly plays a role, but other aspects of a person's identity may be much more influential. For example, the conflict a student perceives between their religious/cultural beliefs and evolution has been shown to be a significant predictor of evolution acceptance [10]. Indeed, as the results from our classroom study seem to indicate, this perceived conflict even has the potential to decrease the agreement with evolution knowledge statements of students who genuinely know the answers to the questions they are being asked.

Rather than increasing this perceived conflict by antagonizing religious beliefs (thereby decreasing student acceptance and even knowledge scores), instructors can improve acceptance in their classrooms and communities by following the Religious Cultural Competence in Evolution Education (ReCCEE) framework [63]. In an effort to improve evolution acceptance among students, the creators of this framework proposed six key practices: (1) acknowledging that students may perceive a conflict, (2) exploring student views on evolution and religion, (3) teaching the nature of science, (4) outlining the spectrum of viewpoints, (5) providing role models, and (6) highlighting potential compatibility. Instructors who have tested culturally competent classroom instruction that mirrors this framework have found great success in reducing perceived conflict and increasing student acceptance of evolution (e.g., [51,46,64,65].

Both the nationwide and classroom studies confirmed previous studies that evolution knowledge is predictive of evolution acceptance (c.f., [6]). These two studies together suggest that perhaps educators and researchers alike need to be sensitive to the idea that religion is a stigmatized identity within scientific communities, meaning that negative stereotypes are attributed to those with religious beliefs, who then in turn may conceal their religious identities or disengage from the learning process in order to avoid these stigmas against holding religious convictions [56]. As a result, religious individuals may be less inclined to participate in a scientific community at all, and those who do often find themselves at odds with factual information about evolution, either in actuality or by perception alone.

Additionally, future survey question wording should be critically analyzed when seeking information about evolution knowledge from Judeo-Christian individuals, thereby avoiding conflating knowledge with acceptance. Adding "according to science" in front of each question, as was done in this study using the EALS instrument with undergraduate Christian students, could help future researchers accurately determine the actual knowledge of the individual taking the survey,

and not a response mitigated by religious belief. These findings emphasize the need to be more precise in the way that we assess evolutionary knowledge in our research studies. As was pointed out in a recent review [66], many instruments commonly used to assess acceptance may perform differently within different religious populations. We would argue that evolution knowledge instruments may suffer from the same flaws.

## Limitations and future research

We collected data from 827 people across the United States. Approximately half of these people identified with a Judeo-Christian religious background and the other half identified as agnostic/atheist. Despite the large sample size, these data cannot be considered generalizable to non-Christian religions since other religious backgrounds were not included in this study.

Our classroom data studied 729 non-majors taking an introductory biology course at a highly religious and religiously homogeneous school. Though these data may not be generalizable among the student population nationally due to the specific population group, they can be representative of college biology students representing a predominantly conservative Christian viewpoint. These findings can be used by researchers and educators alike to guide them in gauging and understanding the actual knowledge of those in their community holding conservative Christian beliefs. Certainly, more research needs to be conducted in this area with a wider representation of student demographics, but these data serve as a template for educators to better understand the knowledge of their Christian students.

Furthermore, our evolution knowledge instrument was a Likert-scale, or opinion-based, instrument, automatically putting it at risk of conflation with belief. It would be interesting to explore this further using an instrument designed to have right/wrong answers. Additionally, it may introduce bias, as seen in our classroom study: perceived conflict between religion and evolution can occur with differences in question wording. Future studies could avoid this conflation by carefully wording questions within a given context. For example, Nehm and colleagues [67] developed the ACORNS instrument that asks respondents to explain how a scientist might think about a phenomenon. This avoids conflation by asking students to think a certain way, similar to how "according to science…" questions might have influenced their frame of mind while answering questions. We suggest further research in understanding and developing evolution knowledge tools to separate possible conflations of evolution acceptance with evolution knowledge.

## Conclusion

In a nationwide survey measuring evolution knowledge, we found data to support the claim that agnostic/atheistic individuals, on average, have a significantly higher knowledge of evolution agreement than those who identify with a Judeo-Christian religion. Upon these discoveries, we conducted a classroom study to determine whether question wording may play a role in this difference by conflating acceptance with knowledge as Judeo-Christian individuals conceal evolution knowledge in order to protect religious identity. Minor but important changes to evolution knowledge instruments could be made to increase measurement validity. More accurately measuring evolution knowledge can help researchers determine areas of evolution about which people genuinely lack understanding from those that they know but do not accept.

## Supporting information

**S1 Table. Confirmatory factor and reliability analysis for survey latent.** All factor loadings are significant at p < .001.
(DOCX)

**S1 File. Survey instrument for nationwide survey.**
(DOCX)

## Author contributions

**Conceptualization:** Jonathan D. Hodson, Jamie L. Jensen.

**Data curation:** Jonathan D. Hodson, Dalton Bourne, Jamie L. Jensen.

**Formal analysis:** Jonathan D. Hodson, Dalton Bourne, Noah Emery, Jamie L. Jensen.

**Funding acquisition:** Jamie L. Jensen.

**Investigation:** Jamie L. Jensen.

**Methodology:** Jamie L. Jensen.

**Project administration:** Jamie L. Jensen.

**Supervision:** Jamie L. Jensen.

**Visualization:** Jamie L. Jensen.

**Writing – original draft:** Jonathan D. Hodson, Dalton Bourne, Noah Emery, Jamie L. Jensen.

**Writing – review & editing:** Jonathan D. Hodson, Dalton Bourne, Noah Emery, Jade B. Sorensen, Andrea Phillips, Jamie L. Jensen.

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
