## [Decision Letter · Decision Letter 0]

6 Jun 2025

Conflation between knowledge and acceptance may contribute to the knowledge gap between Christian and non-religious people

PLOS ONE

Dear Dr. Jensen,

We look forward to receiving your revised manuscript.

Kind regards,

Corey Cook

Academic Editor

PLOS ONE

Journal Requirements:

[Internal University funding]. 

Reviewers' comments:

Reviewer's Responses to Questions

**Comments to the Author**

1. Is the manuscript technically sound, and do the data support the conclusions?

Reviewer #1: Partly

Reviewer #2: Partly

2. Has the statistical analysis been performed appropriately and rigorously?

Reviewer #1: I Don't Know

Reviewer #2: No

3. Have the authors made all data underlying the findings in their manuscript fully available?

Reviewer #1: Yes

Reviewer #2: Yes

4. Is the manuscript presented in an intelligible fashion and written in standard English?

Reviewer #1: Yes

Reviewer #2: Yes

Reviewer #1: The authors present two studies – a national cross-sectional study and a large experimental study – to examine the relationships between evolutionary acceptance, evolutionary knowledge, and religiosity. Together, these studies are interesting and suggest a novel possibility that Christians suppress knowledge about evolution because it is conflated with acceptance of evolution (and suppression reduces potential conflict between knowledge and acceptance). That being said, I don’t feel this study is currently ready for publication. I’m not sure whether its contribution warrants publication at PlosONE in terms of scope of impact, but I gladly defer to the editorial team on this. I certainly would like to see a more robust, revised version published, and I would be happy to review any revisions. I believe the authors can address my comments with revisions, and I think this distinction could contribute to the broader scholarship in this area.

Primary comments:

1. The first study is simple, but I am left wondering why some variables were unmeasured, given the literature review. The authors say that “the first study presented in this manuscript aimed to establish clear relationships between evolution knowledge, religiosity, and evolution acceptance.” (lines 98-99) However, the literature review rightly pointed to several nuanced factors that impact this relationship (understanding of the nature of science, political ideology, perceived conflict between science (or evolutionary theory specifically), and religion (or their religious beliefs specifically)). I don’t quite know how to interpret the findings, given that these things were not included. I’m not convinced these relationships were established clearly because there are clear influences that were not accounted for. (Although I can note that some things should have been measured, it’s not clear that political ideology – which was measured – served any purpose beyond demographic information. I’m also unclear why this choice was made.)

2. Line 115 indicated that scientific reasoning skills were part of the first study. I don’t see this. Scientific reasoning is also mentioned in study 2 in the context of assumptions made about the students' reasoning skills (see line 274). I think this needs to be clarified and addressed.

3. The discussion raises several possibilities about what the phrase “according to science” is doing to raise evolutionary knowledge. This, I think, is the most interesting contribution of this study. It’s also an area where I think more discussion is required; I don’t think the authors have the data to support their claim (as currently made) and I think other possibilities are not yet ruled out. I’ll share some thoughts here for the authors’ consideration (and to potentially dialogue with).

a. The third option, favored by the authors, assumes the lower knowledge scores without the “according to science” are purposeful (line 338). I don’t think the data support this. It could be purposeful, but it could just as easily be unconscious or unintentional. For example, the phrase could activate a different memory schema (e.g., a form of context-dependent memory) that activates the evolutionary knowledge structures in place more strongly. However, for the most religious, this activation may require prompting to override a “default” religious interpretation/knowledge structure. Whether it is intentional or not is a separate empirical question.

b. There was no mention of trust in science or trust in scientists (which don’t seem to operate the same). How might trust in either impact (or interact) with reasoning according to science? Political affiliation (and specific denominational affiliations) might matter here quite a bit. For some, asking “according to science” might not mitigate potential conflict (as mentioned in line 336), but might exacerbate it (e.g., equipping an anti-evolution apologetic structure; reasoning is improved, but the conflict is also widened). Like the above comment, this is an empirical question, and the authors don’t have the data to decide between these options.

c. Asking for responses “according to science” could also activate different identity/social systems. Was there an interaction with gender (a sloppy proxy for empathy)? I would also be interested in ruling out the possibility that asking “according to science” did something with social identity, “reasoning as if I were a scientist” rather than demonstrating stable knowledge. For example, if the participants don’t see themselves as especially science-inclined, reasoning according to science could just mean “the opposite of what I think” leading unknowing individuals into a more correct response than otherwise possible.

4. One limitation discussed was using a Likert scale measure for evolution knowledge. Given this, some intentional attention to the directions would be helpful. Do they understand this as a knowledge test? (The instructions are not included with the question in the instructional materials, but given that the experimental manipulation was with instructions, this is a significant omission.)

Other comments:

1. Were the demographic descriptions of the nationwide sample collected as part of the study, or were these the screener criteria from Qualtrics? This should be clarified (and question wording provided, if relevant). I’m accustomed to more robust participant descriptions and find myself feeling the lack of that in both of these studies.

2. Although the questions to measure religiosity were provided in the supplemental materials, I did not see any statistical validation of these measures in this sample. Since not all of the questions are decidedly religious (e.g., my children will have a better life), this seems important.

3. Although evolution acceptance was lower among religious respondents in the national sample, it seemed higher than I would have expected, given previous research. Is that true? Contextualizing the size of the effects (within this study and relative to previous studies) would be helpful. (This comment also applies to the ~3% increase in the classroom study when asked to answer “according to science” – where did this move reasoning, relative to the atheist sample in the national survey or previous research?)

4. In the national sample, one item for evolutionary acceptance didn’t load for the religious sample, and a different one didn’t load for the atheist/agnostic sample. Why was this? What do you think was happening with this question in these groups? This was never discussed.

5. Although the college sample may be homogenous religiously, the national sample combines huge swaths of “Christian tradition” that may not be best understood as a singular group. (Same thing with atheists/agnostics, too.) I think some attention to these potential differences – and some assurance to your reader that it is right to combine them for these analyses – is important. “Christians” can include denominations that explicitly accept (and endorse) evolution and those that are staunchly against it. (This is even true within the same denomination, with more conservative/liberal factions.) Sometimes getting too detailed in these things can make us lose sight of the forest amidst the trees but combining them without asking these questions first can also obscure important differences.

Reviewer #2: The research article by Hodson et al. addresses a critical issue in evolution education by examining the relationship between evolution acceptance and knowledge, particularly among Christian and non-religious students. The authors explore the potential conflation of these two constructs and investigate how perceived conflict with religious beliefs may influence how students respond to evolution knowledge assessments. This review evaluates the study’s significance, introduction, methodology, results, and conclusions.

Significance of the Study:

This study makes a novel contribution by shifting focus from simply measuring the relationship between evolution knowledge and acceptance to exploring the possibility that these constructs may be conflated, particularly among religious students. While previous research has often compared differences between religious and non-religious students, this study goes further by experimentally probing the effects of perceived conflict on students’ responses by the inclusion of a quasi-experimental classroom component. This methodological innovation provides valuable insight into how religious identity may influence not only evolution acceptance, but also the expression of evolution knowledge. Despite its strengths on significant of the study, there are several aspects that would benefit from clarification and further development. Therefore, I recommend a major revision to strengthen the work.

Introduction:

Lines 50–54. The sentence beginning with “This research has shown…” would benefit from clarification. The pronoun “this” is ambiguous, does it refer to the authors' current study or to the body of prior literature? I suggest the author to name the research explicitly (e.g., “Prior survey research of general U.S. population (Dunk et al., 2017)…”). Also, the claim that “Christian religious affiliation is a major predictor of evolution rejection” should be more carefully framed. While Dunk et al. (2017) did find a statistically significant relationship between religious denomination and evolution acceptance using a general linear model, the effect size reported suggests a small effect. Although the religious denomination contributes to the variance, but it may not be appropriate to characterize it as a “major” predictor without further qualification. The authors could strengthen this section by explicitly reporting the size and context of the effect.

Lines 57–59. The authors summarize that “affiliation with and commitment to Christian, or biblically grounded, religious belief is likely to hinder the acceptance of evolution.” This conclusion, while supported by survey data such as the Pew reports, would benefit from contextualization. Since the focus here is on data from the United States, it would be important to reiterate that these trends are culturally and geographically situated, and may not generalize to global patterns of evolution acceptance.

Lines 60–78 (2nd–3rd paragraphs). Before discussing into the complex relationship between religion and evolution, the introduction would benefit from a clear and early articulation of how the authors conceptualize and operationalize “evolution acceptance” and “evolution rejection.” Referring to existing literature that defines these constructs, such as Barnes et al. (2021) that reviews the validity of current evolution acceptance instruments, would add conceptual clarity and strengthen the theoretical framing. Additionally, Lines 63-66, it would be useful if the authors specified which dimensions of religious belief (e.g., belief in God, religious teaching, religious community) are most significant conflict predictor of evolution acceptance.

Lines 79–94 (4th paragraph). Prior work that models’ interactions between knowledge and religiosity is omitted. The author could cite U.S. public-sample research (Weisberg et al., 2018) and student-sample work (Aini et al., 2024) showing that knowledge predicts acceptance more strongly among individuals with lower religiosity. This literature better supports author’s argument that the knowledge–acceptance link is conditionally moderated by religious variables.

Lines 95–111. While this section is a strength especially stating the hypothesis that knowledge scores may be conflated with acceptance is clearly articulated, there are major area of improvement especially in the terms of structure sentences. In some sentences, it blurs the boundary between the introduction of hypotheses and discussion of supporting evidence. Rather than citing previous studies as support for the hypothesis at this stage (e.g., Mead et al., 2018; Winslow et al., 2011), it would be more appropriate to present those findings as part of the rationale for the current research. This would avoid giving the impression that the hypothesis has already been confirmed. Reorganizing this section to clearly separate the hypothesis from the discussion of prior findings would enhance clarity and maintain a logical flow.

Lines 112–115. The sentence describing the first study (“the first study examined these relationships on a large scale through administration of a survey targeting evolution knowledge, evolution acceptance, scientific reasoning skills, and religiosity”) feels more appropriate for the Methods section. It can be confusing for the readers since it introduces "scientific reasoning skills" as a variable of interest, even though earlier in the introduction the authors seem to downplay or dismiss its relevance to evolution acceptance.

Methods:

Lines 135–137. While attention checks are useful for improving data quality, they do not constitute formal validity evidence in a psychometric sense (e.g., content, construct, or criterion validity). It would be more appropriate to revise this sentence.

Line 137. Information about the response rate is missing. Including this would help readers assess the potential for nonresponse bias, which can impact the generalizability and representativeness of the findings.

Table 1 / Line 143. The categorization of participants as “Theist (Judeo-Christian)” is unclear and potentially problematic. The term “Christian” was consistently used in the introduction, and it would be more informative to retain consistency and specify denominational affiliations. This distinction is important given that prior research (e.g., Dunk et al., 2017) has shown denominational differences in evolution acceptance. Clarifying this terminology will enhance interpretability and theoretical alignment with the literature discussed earlier.

Lines 162–165. The authors report using a subscale from the Evolution Attitudes and Literacy Survey–Short Form (EALS-SF); however, more detail is needed. It is unclear which subscale was used and how many items were included. Providing example items would allow readers to more effectively evaluate the appropriateness of the measure in relation to the study's objectives.

A conceptual concern with this instrument should also be addressed. While the stated aim of the study is to examine whether evolution knowledge and acceptance are being conflated in student responses, the nature of the EALS-SF itself may contribute to this conflation. Specifically, the use of a Likert-type agreement scale to assess knowledge risks capturing attitudinal agreement rather than cognitive understanding. This blending of epistemic and affective constructs is especially problematic in a study focused on disentangling the two.

The authors should also consider discussing the consequential validity of the EALS-SF, especially the implications of interpreting agreement scores as indicators of a student's knowledge, comprehension, or application of evolutionary theory, rather than as expressions of their acceptance of its scientific validity. This concern is central to the paper’s argument and should be explicitly acknowledged, ideally both in the introduction and in the interpretation of results.

Line 186-190. This sentences that describes the structure and scoring of the evolution knowledge instrument, would be more appropriately placed in the earlier section where the evolution knowledge measure is first introduced since it’s important details from the description of the instrument itself.

Line 194: The authors should specify the estimator used in the confirmatory factor analysis (e.g., maximum likelihood, weighted least squares). The choice of estimator has important implications for model fit and the treatment of ordinal data, especially when Likert-scale responses are used.

Results

Lines 218–221: The authors report that participants identifying as agnostic or atheist scored significantly higher on the evolution knowledge assessment than participants identifying with Judeo-Christian religions. The terminology used across the manuscript lacks consistency. In the introduction and figures, the groups are labeled as “atheist/agnostic” and “religious,” but here the text specifies “Judeo-Christian” in the method or “Christian” in the rest of manuscript. For clarity and alignment across sections, the authors should consistently define and use their grouping terms—e.g., “religious vs. nonreligious,” “Christian vs. atheist/agnostic,” or “Judeo-Christian vs. atheist/agnostic”—and specify how these categories were operationalized. Additionally, consider explaining the rationale for grouping atheists and agnostics together and distinguishing them from religious participants, as these groups may differ in epistemological

views.

Lines 249–253: The authors’ interpretation of predictive power based on standardized factor loadings in the structural equation model is problematic. Factor loadings describe the relationship between latent constructs and their observed indicators and are not valid indicators of predictive strength. Predictive power should be evaluated using standardized path coefficients and the amount of variance explained (e.g., R² values) in the dependent variable. Even if the authors were referring to standardized regression paths, direct comparison of path coefficients across groups requires multi-group SEM with formal testing procedures (e.g., chi-square difference tests with equality constraints). Without this statistical support, the conclusion about differences in predictive power across groups should be interpreted cautiously or revised entirely.

Discussion:

Lines 269–271. The discussion begins by stating that agnostic/atheist respondents agreed more with evolution knowledge items than Christian respondents. Again, the inconsistent use of group labels (e.g., “Christian,” “Judeo-Christian,” “religious,” “agnostic/atheist,” “nonreligious”) may confuse readers and should be standardized throughout the paper. The authors should clearly define each group and justify the classification scheme.

Lines 279–282: This section of the discussion effectively highlights how inserting the phrase “According to science…” into evolution knowledge items resulted in higher scores, even though the item content remained unchanged. This finding provides compelling evidence for the central claim of the paper, that student responses to knowledge assessments may be influenced by their acceptance of evolution, revealing a possible conflation between conceptual understanding and personal belief. To strengthen this point, the authors should offer a more explicit critique of the instrument used. Specifically, the EALS assesses "evolutionary knowledge" using agreement-based Likert scales, which inherently risk conflating acceptance with understanding. This measurement approach may not fully separate students’ conceptual grasp of evolution from their attitudinal alignment with the statements. To support this critique, the authors could cite studies that have attempted to disentangle knowledge from acceptance using alternative epistemic framings. For instance, the ACORNS instrument (Nehm et al., 2012) prompts students to explain how a biologist might account for evolutionary phenomena, thereby framing responses explicitly within a scientific context. Research using such instruments has shown that question wording and framing significantly affect how students—particularly those experiencing perceived conflict between science and religion—respond to evolution-related assessments.

Lines 286–316 (Three Explanatory Hypotheses). The authors present three potential explanations for the observed lower knowledge scores among religious students. First, Conflict-induced identity-protective cognition (Kahan et al., 2007), second Stigmatization and disengagement, and last Measurement-induced conflation (students masking understanding due to religious conflict). While the third explanation is central to the study’s contribution, the first two explanations are only loosely connected to the findings and are not empirically supported within this study. These cognitive and affective frameworks would be more appropriately discussed in the introduction, where they could help establish the theoretical foundation for the study’s hypothesis. In contrast, the third explanation—the potential conflation of knowledge and acceptance in survey responses—is directly supported by the study’s findings and should be emphasized more strongly in the discussion.

Lastly, Given the methodological implications of this study, it would be helpful to include a limitations section directly following the Methods. For example, concerns about the psychometric validity of the EALS-SF (e.g., use of Likert scales to assess knowledge), the absence of formal invariance testing in SEM comparisons, and the potential role of social desirability or response bias should be acknowledged upfront. This would allow readers to interpret the results and conclusions within an appropriately cautious framework.

**Do you want your identity to be public for this peer review?** For information about this choice, including consent withdrawal, please see our Privacy Policy

Reviewer #1: No

Reviewer #2: **Yes: ** Rahmi Aini

---

## [Author Response · Author response to Decision Letter 1]

28 Jul 2025

(Please see attached file for better formatting)

Reviewer Comment Response to Reviewer

Reviewer #1: The authors present two studies – a national cross-sectional study and a large experimental study – to examine the relationships between evolutionary acceptance, evolutionary knowledge, and religiosity. Together, these studies are interesting and suggest a novel possibility that Christians suppress knowledge about evolution because it is conflated with acceptance of evolution (and suppression reduces potential conflict between knowledge and acceptance). That being said, I don’t feel this study is currently ready for publication. I’m not sure whether its contribution warrants publication at PlosONE in terms of scope of impact, but I gladly defer to the editorial team on this. I certainly would like to see a more robust, revised version published, and I would be happy to review any revisions. I believe the authors can address my comments with revisions, and I think this distinction could contribute to the broader scholarship in this area. Thank you. We hope that the revisions have made the paper robust enough to warrant publication

1. The first study is simple, but I am left wondering why some variables were unmeasured, given the literature review. The authors say that “the first study presented in this manuscript aimed to establish clear relationships between evolution knowledge, religiosity, and evolution acceptance.” (lines 98-99) However, the literature review rightly pointed to several nuanced factors that impact this relationship (understanding of the nature of science, political ideology, perceived conflict between science (or evolutionary theory specifically), and religion (or their religious beliefs specifically)). I don’t quite know how to interpret the findings, given that these things were not included. I’m not convinced these relationships were established clearly because there are clear influences that were not accounted for. (Although I can note that some things should have been measured, it’s not clear that political ideology – which was measured – served any purpose beyond demographic information. I’m also unclear why this choice was made.) Thank you for pointing out this ommision. We actually did gather quite a bit more data and in our efforts to simplify the findings, we inadvertantly omitted important information. We did measure scientific reasoning ability, religiosity, and political ideology. We did not measure perceived conflict as the pCORE (a great instrument for measuring this conflict) was not published when we gathered these data. We did include a few questions from the EALS that sort of gets at conflict but they proved to be rather unreliable and didn't make sense to include in our analysis. We did not measure nature of science understanding, although we wish we had. We have gone back and re-run analyses including the religiosity, ideology, and education level variables and I do believe it significantly adds to the depth of these relationships. We did not, however, include the scientific reasoning test (see next comment). We appreciate the reviewer for pointing this out.

2. Line 115 indicated that scientific reasoning skills were part of the first study. I don’t see this. Scientific reasoning is also mentioned in study 2 in the context of assumptions made about the students' reasoning skills (see line 274). I think this needs to be clarified and addressed. While we did attempt to measure scientific reasoning ability (just to confirm previous findings of no relationship), we ultimately decided to omit it from analyses due to validity issues. We have briefly stated this in the paper, but we are happy to provide you with additional details here: First, the first four items were non-descriminatory because every respondent got them correct. (This was worrisome in and of itself given that it is unusal for all respondents to get all correct, so we may have had a technical error here). An additional item (item 12) had all respondents getting it wrong. Additionally, the correlational reasoning construct was only measured by two items (this was how the original instrument was designed) and thus could not be included in a CFA if we parsed it out be reasoning abilty as factors. Additionally, two of the four items on the Hypothetico-deductive construct (which, incidentally is an important construct that distinguishes formal and post-formal reasoners from concrete reasoners), showed responses that were no better than guessing and we worried that respondents encountered these longer and more complicated items and they did not try. Lastly, we looked at the average scores on the reasoning test (including all 24 items) by educational level and found that the average ranged from 10 to 14 out of 24. In previous work that we have done (and others; see Jensen, J.L., et al. 2018. Investigating strategies for pre-class content learning in a flipped classroom, Journal of science Education and Technology, 27, 523-535; Lawson, A. E. et al., 2000. What kinds of scientific concepts exist? Concept construction and intellectual development in college biology, Journal of Research in Science Teaching, 37, 996-1018; Jensen, J. L. et al., 2015, Learning scientific reasoning skills may be key to retention in science, technology, engineering, and mathematics, Journal of College Student Retention: Research, Theory, & Practice, 0, 1-19.), we have found these levels (especially for those with advanced degrees) to be far lower than we would expect, again leading us to believe that participants may have just guessed on these more complicated items.

3. The discussion raises several possibilities about what the phrase “according to science” is doing to raise evolutionary knowledge. This, I think, is the most interesting contribution of this study. It’s also an area where I think more discussion is required; I don’t think the authors have the data to support their claim (as currently made) and I think other possibilities are not yet ruled out. I’ll share some thoughts here for the authors’ consideration (and to potentially dialogue with).

a. The third option, favored by the authors, assumes the lower knowledge scores without the “according to science” are purposeful (line 338). I don’t think the data support this. It could be purposeful, but it could just as easily be unconscious or unintentional. For example, the phrase could activate a different memory schema (e.g., a form of context-dependent memory) that activates the evolutionary knowledge structures in place more strongly. However, for the most religious, this activation may require prompting to override a “default” religious interpretation/knowledge structure. Whether it is intentional or not is a separate empirical question. This is a fascinating idea and one that we had not considered! We have added to our discussion this possibility along with the possibilitiy suggested in "b". Very inciteful.

b. There was no mention of trust in science or trust in scientists (which don’t seem to operate the same). How might trust in either impact (or interact) with reasoning according to science? Political affiliation (and specific denominational affiliations) might matter here quite a bit. For some, asking “according to science” might not mitigate potential conflict (as mentioned in line 336), but might exacerbate it (e.g., equipping an anti-evolution apologetic structure; reasoning is improved, but the conflict is also widened). Like the above comment, this is an empirical question, and the authors don’t have the data to decide between these options. Thank you for this suggestion. We have expanded our discussion. Unfortunately, we do not have political affilitation for these students (and almost all would have been affiliated with the CJCLDS church, so there are no differences there). But, it is interesting to suggest that it perhaps prompted a negative affect (as suggested in c). We have included this possibility. Brilliant!

c. Asking for responses “according to science” could also activate different identity/social systems. Was there an interaction with gender (a sloppy proxy for empathy)? I would also be interested in ruling out the possibility that asking “according to science” did something with social identity, “reasoning as if I were a scientist” rather than demonstrating stable knowledge. For example, if the participants don’t see themselves as especially science-inclined, reasoning according to science could just mean “the opposite of what I think” leading unknowing individuals into a more correct response than otherwise possible. We unfortunately do not have any identifying information for these students beyond the aggregate student demographics mentioned in the paper. The survey was purposefully anonymous. But, I do like the possibility of "the opposite of what I think" playin a role. We have included that.

4. One limitation discussed was using a Likert scale measure for evolution knowledge. Given this, some intentional attention to the directions would be helpful. Do they understand this as a knowledge test? (The instructions are not included with the question in the instructional materials, but given that the experimental manipulation was with instructions, this is a significant omission.) Great catch. There was no specific instruction preceding the knowledge questions that would look any different than the acceptance questions. We agree that this is very important to point out. It further strengthens the need to be precise in our instrumentation when we measure "knowledge of evolution". We have added this detail to our methods section (and included the full survey in the supplement).

Other comments:

1. Were the demographic descriptions of the nationwide sample collected as part of the study, or were these the screener criteria from Qualtrics? This should be clarified (and question wording provided, if relevant). I’m accustomed to more robust participant descriptions and find myself feeling the lack of that in both of these studies. They were collected as part of the survey (rather than as screening). We have clarified that in the manuscript. And, great catch! We should have included the full survey in the supplement. We have done that now so wording can be assessed.

2. Although the questions to measure religiosity were provided in the supplemental materials, I did not see any statistical validation of these measures in this sample. Since not all of the questions are decidedly religious (e.g., my children will have a better life), this seems important. We apologize for the oversight. We have amended the supplement to include both the full survey and the full CFA statistics for each measure, including religiosity.

3. Although evolution acceptance was lower among religious respondents in the national sample, it seemed higher than I would have expected, given previous research. Is that true? Contextualizing the size of the effects (within this study and relative to previous studies) would be helpful. (This comment also applies to the ~3% increase in the classroom study when asked to answer “according to science” – where did this move reasoning, relative to the atheist sample in the national survey or previous research?) Regarding evolution acceptance at the nationwide level, I don't believe the numbers are surprising. Accroding to poll data we cited in the intro for contextualization, nationwide acceptance is around 60% for the general public, and closer to 82% if you include a theistic evolution viewpoint. And amongst Christians, that number ranges between 42 and 64% when just looking at human evolution. We got an average of 96 on a 138-point scale (given the low is 23), which included micro, marco, and human evolution. This would indicate a value very close to what we are seeing nationwide. However, regarding the 3% increase in the classroom study, this is a valid point to consider. It is a small increase, to be sure, but it was also a very small intervention. It is difficult to contextualize this given that other studies use different instruments, so I'm unsure as to how to compare it to others, as well as the fact that we did an entirely new intervention that has not been done.

4. In the national sample, one item for evolutionary acceptance didn’t load for the religious sample, and a different one didn’t load for the atheist/agnostic sample. Why was this? What do you think was happening with this question in these groups? This was never discussed. This is a good question. Unfortunately, looking at the two items, we can't really come up with an explanation to adequately explain why these would be different between religious and non-religious populations. We suspect it is just due to random messiness in the data, as the fit was very close on each item. We are not sure that a discussion of this would be helpful to the reader.

5. Although the college sample may be homogenous religiously, the national sample combines huge swaths of “Christian tradition” that may not be best understood as a singular group. (Same thing with atheists/agnostics, too.) I think some attention to these potential differences – and some assurance to your reader that it is right to combine them for these analyses – is important. “Christians” can include denominations that explicitly accept (and endorse) evolution and those that are staunchly against it. (This is even true within the same denomination, with more conservative/liberal factions.) Sometimes getting too detailed in these things can make us lose sight of the forest amidst the trees but combining them without asking these questions first can also obscure important differences. this is an excellent point. We have run analyses for each outcome variable separately between religious affiliations and between agnostic/atheist respondents. There are differences that we have included in a descriptive Table (Table 2). Many of the religious groups do not have a representative sample to draw any definitive conclusions about them. We eliminated all religions that had 10 or fewer respondents (a natural break in the data) and re-ran all analyses to see if the patterns changed significantly. They did not, so we opted to maintain all the data. However, we have made notes on this throughout the results, and included Table 2 so people can really assess the data in context. We appreciate the suggestion.

Reviewer #2: The research article by Hodson et al. addresses a critical issue in evolution education by examining the relationship between evolution acceptance and knowledge, particularly among Christian and non-religious students. The authors explore the potential conflation of these two constructs and investigate how perceived conflict with religious beliefs may influence how students respond to evolution knowledge assessments. This review evaluates the study’s significance, introduction, methodology, results, and conclusions.

Significance of the Study:

This study makes a novel contribution by shifting focus from simply measuring the relationship between evolution knowledge and acceptance to exploring the possibility that these constructs may be conflated, particularly among religious students. While previous research has often compared differences between religious and non-religious students, this study goes further by experimentally probing the effects of perceived conflict on students’ responses by the inclusion of a quasi-experimental classroom component. This methodological innovation provides valuable insight into how religious identity may influence not only evolution acceptance, but also the expression of evolution knowledge. Despite its strengths on significant of the study, there are several aspects that would benefit from clarification and further development. Therefore, I recommend a major revision to strengthen the work. Thank you, we hope our revisions provided the clarification you were hoping to see.

Introduction:

Lines 50–54. The sentence beginning with “This research has shown…” would benefit from clarification. The pronoun “this” is ambiguous, does it refer to the authors' current study or to the body of prior literature? I suggest the author to name the research explicitly (e.g., “Prior survey research of general U.S. population (Dunk et al., 2017)…”). Also, the clai

---

## [Decision Letter · Decision Letter 1]

23 Sep 2025

Dear Dr. Jensen,

Thank you for submitting your revised manuscript to PLOS ONE. Each of the original reviewers agreed to review your revised manuscript, and each offer helpful recommendations to improve the clarity of the research. Therefore, we invite you to submit a revised version of the manuscript that addresses the points raised during the review process. I am particularly interested in Reviewer 2's questions regarding the interpretive framing of the results. 

We look forward to receiving your revised manuscript.

Kind regards,

Corey Cook

Academic Editor

PLOS ONE

Journal Requirements:

Reviewers' comments:

Reviewer's Responses to Questions

**Comments to the Author**

Reviewer #1: (No Response)

Reviewer #2: All comments have been addressed

2. Is the manuscript technically sound, and do the data support the conclusions?

Reviewer #1: Yes

Reviewer #2: Partly

3. Has the statistical analysis been performed appropriately and rigorously?

Reviewer #1: I Don't Know

Reviewer #2: No

4. Have the authors made all data underlying the findings in their manuscript fully available?

Reviewer #1: Yes

Reviewer #2: No

5. Is the manuscript presented in an intelligible fashion and written in standard English?

Reviewer #1: Yes

Reviewer #2: Yes

Reviewer #1: I am grateful to review this revised manuscript. It clarifies many ambiguities and presents a more streamlined presentation. There are, however, a few questions I'll raise, in hopes of encouraging even more clarity. I consider all but the first comment minor.

1. Paper organization. As with the first, both the national sample and the classroom sample are presented as one study. I think this does more harm than good for clarity of communication. I would strongly encourage a Study 1/Study 2 presentation, each with their own (brief) literature review, hypotheses/building rationale, method, analysis, etc. In some cases, a simple, "The same measures were used in the classroom study as in the national study" would suffice. This change will reduce the cognitive load required of the reader to hold all the various pieces in mind, and it will help clarify the value of each part of the paper. For example, in the results, all of the SEM models - were they also run for the classroom study? Why not? If not, why bother give all the students all those measures? Having them separated will allow the authors to more carefully communicate what each study does; what questions they were designed to answer and what they add. Some of the things can be moved to supplemental materials (e.g., the same SEM analyses confirmed similar patterns, see supplemental for full statistical read outs), while leaving the important pieces in the paper narrative. This comment is less about changing content, and more about increasing the ease of engaging that content.

2. Nature of Science. This is still mentioned in the abstract (first sentence!) and in the literature review. Since it wasn't measured, it seems odd to draw attention to it in these places. It feels like a self-inflicted loose end! Help me, as your reader, situate the research on NoS within your study, even if it isn't measured. Could identity (which you discuss) be a gateway to NoS understanding? (I'm thinking of the research on the #ilooklikeanengineer campaign, though I think other work likely applies.)

3. Table 1. Are the demographics different by religious affiliation? This shows up later in the manuscript (lines 320-326), but it seems they should be reported here for clarity and consistency.

4. Survey measures. Were the surveys given in a set order (both studies) or were they randomized? Can priming effects be ruled out?

5. Figure 3 (and associated analysis). Is there a relationship between religious influence and knowledge of evolution? That doesn't seem to be accounted for in the model.

6. Speculation. This wouldn't need to be addressed, though I'll share this comment because I think a marker of good research is work that invites new questions: Related to #5, what do you think the relationship between religious influence and knowledge of evolution ought to be (longitudinally)? The answer to this question might actually help sort out some of the possibilities posed in your paper. If it is primarily social identity at play (vs. cognitive dissonance or "science as encouraging deconversion"), then would we expect religious influence  knowledge (vs. knowledge  religious influence)? These aren't necessarily mutually exclusive influences, but I wonder if more attention could be paid to contextualizing these findings in light of bigger questions around religion, sociocultural context, belief, and knowledge. As I said, this could be addressed, though I wouldn't expect it to be given the particular focus/restrictions, etc.

Reviewer #2: I still to believe that this study has the potential to make a novel contribution, particularly by providing evidence on how the relationship between knowledge/understanding and acceptance can be conflated. It also helps shift the conversation from a deficit model toward exploring the psychological processes that shape how knowledge is expressed by religious students.

Despite the strengths of the study and my appreciation for the revisions made, I believe further clarity is needed in the methodology. I remain especially concerned about how the measurement was treated and how the scores were interpreted. I am not yet convinced that the current analytical approach supports the claims being made.

Sampling procedure clarity.

More detail is needed about the sampling procedures for the nationwide study. The manuscript notes the use of the Qualtrics surveying platform, but it is unclear whether Qualtrics was only used to host and distribute the survey or whether only recruitment occurred through Qualtrics. How were participants recruited nationwide through Qualtrics? Without this information, it is difficult to assess the representativeness of the sample, as claimed by the author.

Instrument clarity.

There is still inconsistency in the description of the instrument. Figures 2 and 3 suggest that four EALS items were used, yet the supplemental materials list eight items (including items from the “misconceptions about evolution” subscale), and line 197 refers to five items. This inconsistency makes it unclear what was actually included in the analyses.

Level of measurement and score interpretation

I appreciate the authors’ revisions acknowledging that the EALS-SF subscale is opinion-based and therefore at risk of conflation with belief. However, several important concerns remain unresolved, particularly regarding score interpretation and its statistical and conceptual implications.

The paper’s central claim is that measured “evolution knowledge” differs across religious groups and how it predicts acceptance. The instrument, however, is based on Likert-type agreement items that were summed into total scores (as described in the methods and statistical results) but sometimes expressed as percentages (as shown in Figure 1 and in the discussion). While using summed Likert scores in t-tests is common practice in our field, especially with multiple items and large samples, it still assumes interval-level properties that are not strictly supported by ordinal data. My greater concern arises from converting these scores into percentages, which suggests ratio-level meaning that the instrument cannot support. For example, the authors reported a “57% knowledge score” in the discussion, which implies a proportion of correct knowledge, when in reality the value represents aggregated agreement ratings. This creates a risk of over-interpreting what the measure actually captures and may mislead readers into equating endorsement with factual knowledge.

To give a concrete example, the finding that adding the phrase “according to science” revealed a divergence between the summed scores for “knowledge” and acceptance is one of the most interesting results of the study. The interpretation, that religious students may filter their responses through a schema tied to religious identity, either deliberately or unconsciously, is important. However, the way the scores are reported risks overstating what the instrument actually captures. For instance, describing a student as having “26% on acceptance but 83% on knowledge” implies a ratio-level interpretation that the measure cannot support. These percentages do not represent proportions of correct knowledge; again they are re-scaled aggregates of ordinal agreement ratings. The value of the classroom study lies in showing that the relationship between acceptance and knowledge scores changes under different framings, but the exact percentages (e.g., 26% vs. 83%) are less meaningful, because they suggest a level of precision and factual correctness that the instrument does not provide. Reporting summed or average scores, providing the statistical relationship, and explicitly cautioning readers about what these values represent would strengthen the evidentiary basis for this interpretation.

In other words, while the qualitative pattern observed (greater divergence under the “according to science” framing) is meaningful, the magnitude and precision implied by percentage scores and statistical tests need further consideration, especially since the instrument was not designed with right/wrong answers or validated for interval-level analysis.

Interpretive framing

I wonder if using average Likert scores or consistently using summed scores rather than percentages would change the results. If so, this would suggest that using percentages is indeed overstating the findings. The authors might also reconsider whether the construct is better framed as “evolution understanding” or “perceived understanding of evolution,” rather than “knowledge of evolution.” This would make clearer that the instrument captures willingness to endorse science-framed statements, not factual correctness. Such a reframing would also align more closely with the authors’ psychological interpretation that students may possess knowledge but choose whether or not to reveal it depending on context.

Alternative analyses

To further strengthen the analysis, I suggest presenting the distribution of responses across Likert categories for the control versus “according to science” groups, rather than reducing them to percentages. This would more directly illustrate how framing shifts patterns of agreement. The authors could also consider statistical models that respect the ordinal nature of the data (e.g., cumulative logit models), which avoid over-interpreting the numeric distances between Likert points. Finally, scatterplots or density plots could be used to visualize how acceptance and understanding scores relate to one another in the two conditions. This would highlight the divergence without relying on potentially misleading percentage-based claims.

**Do you want your identity to be public for this peer review?** For information about this choice, including consent withdrawal, please see our Privacy Policy

Reviewer #1: No

Reviewer #2: No

---

## [Author Response · Author response to Decision Letter 2]

28 Oct 2025

Reviewer #1: I am grateful to review this revised manuscript. It clarifies many ambiguities and presents a more streamlined presentation. There are, however, a few questions I'll raise, in hopes of encouraging even more clarity. I consider all but the first comment minor.

1. Paper organization. As with the first, both the national sample and the classroom sample are presented as one study. I think this does more harm than good for clarity of communication. I would strongly encourage a Study 1/Study 2 presentation, each with their own (brief) literature review, hypotheses/building rationale, method, analysis, etc. In some cases, a simple, "The same measures were used in the classroom study as in the national study" would suffice. This change will reduce the cognitive load required of the reader to hold all the various pieces in mind, and it will help clarify the value of each part of the paper. For example, in the results, all of the SEM models - were they also run for the classroom study? Why not? If not, why bother give all the students all those measures? Having them separated will allow the authors to more carefully communicate what each study does; what questions they were designed to answer and what they add. Some of the things can be moved to supplemental materials (e.g., the same SEM analyses confirmed similar patterns, see supplemental for full statistical read outs), while leaving the important pieces in the paper narrative. This comment is less about changing content, and more about increasing the ease of engaging that content.

We really loved this suggestion and we have split the manuscript into two “studies” for ease of reading. Thank you for suggesting this.

2. Nature of Science. This is still mentioned in the abstract (first sentence!) and in the literature review. Since it wasn't measured, it seems odd to draw attention to it in these places. It feels like a self-inflicted loose end! Help me, as your reader, situate the research on NoS within your study, even if it isn't measured. Could identity (which you discuss) be a gateway to NoS understanding? (I'm thinking of the research on the #ilooklikeanengineer campaign, though I think other work likely applies.)

Excellent catch. This was simply an error in editing on our part. In a larger study, we did include measures of scientific reasoning, conflict, and NOS, but those were not included in this study. It should have been removed from the abstract. We did highlight the past findings on the relationship between NOS and evolution acceptance by way of background, but we have reduced the emphasis on this factor given, as you appropriately pointed out, that we did not measure this.

3. Table 1. Are the demographics different by religious affiliation? This shows up later in the manuscript (lines 320-326), but it seems they should be reported here for clarity and consistency.

Agreed. This is a great suggestion. We have now presented the demographic table divided by religious affiliation.

4. Survey measures. Were the surveys given in a set order (both studies) or were they randomized? Can priming effects be ruled out?

The surveys (in each study) were all administered in the same order: Nationwide – demographics, scientific reasoning, religiosity, I-SEA, EALS-SF; Classroom – the I-SEA and the EALS-SF were given in two separate surveys at two different times. I’m not sure what priming effects there would be and could only speculate. However, we have included this ordering for the nationwide sample, just so the reader is aware and can hypothesize about potential effects.

5. Figure 3 (and associated analysis). Is there a relationship between religious influence and knowledge of evolution? That doesn't seem to be accounted for in the model.

There was no significant relationship between religious influence and knowledge (at least among Theist respondents – we did not measure religious influence among atheist/agnostic respondents), so we did not include this relationship. We included a short statement on this lack of relationship in the results.

6. Speculation. This wouldn't need to be addressed, though I'll share this comment because I think a marker of good research is work that invites new questions: Related to #5, what do you think the relationship between religious influence and knowledge of evolution ought to be (longitudinally)? The answer to this question might actually help sort out some of the possibilities posed in your paper. If it is primarily social identity at play (vs. cognitive dissonance or "science as encouraging deconversion"), then would we expect religious influence  knowledge (vs. knowledge  religious influence)? These aren't necessarily mutually exclusive influences, but I wonder if more attention could be paid to contextualizing these findings in light of bigger questions around religion, sociocultural context, belief, and knowledge. As I said, this could be addressed, though I wouldn't expect it to be given the particular focus/restrictions, etc.

It is an interesting question. However, given that we did not see a relationship between religious influence and knowledge, this particular measure is not super informative. However, I think what you are hinting at, and what we seem to be seeing in our paper (that identity as a religious person is driving the lower knowledge) is a fair point. Unfortunately, I don’t think that our particular measure of “Religious Influence” is appropriate to measure this. Our instrument does measure how much religion influences daily life, but in terms of dress and eating habits, rather than decision making. I would guess that, had we thought of it or had they been available at the time, a better measure would have been the pCORE (Barnes et al., 2021) or the pFEAR (Ferguson et al., 2024) as these get more at the perceived conflict, or at the influence of one’s religion in decision making on controversial topics.

Reviewer #2: I still to believe that this study has the potential to make a novel contribution, particularly by providing evidence on how the relationship between knowledge/understanding and acceptance can be conflated. It also helps shift the conversation from a deficit model toward exploring the psychological processes that shape how knowledge is expressed by religious students.

Despite the strengths of the study and my appreciation for the revisions made, I believe further clarity is needed in the methodology. I remain especially concerned about how the measurement was treated and how the scores were interpreted. I am not yet convinced that the current analytical approach supports the claims being made.

Sampling procedure clarity.

More detail is needed about the sampling procedures for the nationwide study. The manuscript notes the use of the Qualtrics surveying platform, but it is unclear whether Qualtrics was only used to host and distribute the survey or whether only recruitment occurred through Qualtrics. How were participants recruited nationwide through Qualtrics? Without this information, it is difficult to assess the representativeness of the sample, as claimed by the author.

Thank you for pointing this omission out. We have further clarified the sampling procedure and the filters used to get a representative sample.

Instrument clarity.

There is still inconsistency in the description of the instrument. Figures 2 and 3 suggest that four EALS items were used, yet the supplemental materials list eight items (including items from the “misconceptions about evolution” subscale), and line 197 refers to five items. This inconsistency makes it unclear what was actually included in the analyses.

Great catch! During the first revision, based on other comments, we did end up removing the first item from the EALS for poor fit. We failed to update our methods section and our supplement. It has been updated now to indicate that we used only four items. We have left it in the Supplemental File of the original survey, since the original survey included all five and included the “misconceptions about evolution” that were not used in this study.

Level of measurement and score interpretation

I appreciate the authors’ revisions acknowledging that the EALS-SF subscale is opinion-based and therefore at risk of conflation with belief. However, several important concerns remain unresolved, particularly regarding score interpretation and its statistical and conceptual implications.

The paper’s central claim is that measured “evolution knowledge” differs across religious groups and how it predicts acceptance. The instrument, however, is based on Likert-type agreement items that were summed into total scores (as described in the methods and statistical results) but sometimes expressed as percentages (as shown in Figure 1 and in the discussion). While using summed Likert scores in t-tests is common practice in our field, especially with multiple items and large samples, it still assumes interval-level properties that are not strictly supported by ordinal data. My greater concern arises from converting these scores into percentages, which suggests ratio-level meaning that the instrument cannot support. For example, the authors reported a “57% knowledge score” in the discussion, which implies a proportion of correct knowledge, when in reality the value represents aggregated agreement ratings. This creates a risk of over-interpreting what the measure actually captures and may mislead readers into equating endorsement with factual knowledge.

To give a concrete example, the finding that adding the phrase “according to science” revealed a divergence between the summed scores for “knowledge” and acceptance is one of the most interesting results of the study. The interpretation, that religious students may filter their responses through a schema tied to religious identity, either deliberately or unconsciously, is important. However, the way the scores are reported risks overstating what the instrument actually captures. For instance, describing a student as having “26% on acceptance but 83% on knowledge” implies a ratio-level interpretation that the measure cannot support. These percentages do not represent proportions of correct knowledge; again they are re-scaled aggregates of ordinal agreement ratings. The value of the classroom study lies in showing that the relationship between acceptance and knowledge scores changes under different framings, but the exact percentages (e.g., 26% vs. 83%) are less meaningful, because they suggest a level of precision and factual correctness that the instrument does not provide. Reporting summed or average scores, providing the statistical relationship, and explicitly cautioning readers about what these values represent would strengthen the evidentiary basis for this interpretation.

In other words, while the qualitative pattern observed (greater divergence under the “according to science” framing) is meaningful, the magnitude and precision implied by percentage scores and statistical tests need further consideration, especially since the instrument was not designed with right/wrong answers or validated for interval-level analysis.

We agree that this is a tricky instrument. We had opted for percentages to make it easier for the reader to interpret. However, we re-ran statistics using summed scores and, as expected, the statistics are exactly the same. So, I don’t think percentages are overstating the difference. However, we agree with you that percentages are probably not appropriate for Likert-style data. It’s always tricky to convey Likert scores well and make them easily interpretable given the ambiguous distance between ratings. So, we thought that perhaps including a stacked bar of the Likert distributions would at least allow the reader to interpret the raw summed scores a little better (see Figure 4).

We also agree that the framing was a little funky, as well. We have gone through and reframed the results to reflect a better representation of Likert-style data.

Interpretive framing

I wonder if using average Likert scores or consistently using summed scores rather than percentages would change the results. If so, this would suggest that using percentages is indeed overstating the findings. The authors might also reconsider whether the construct is better framed as “evolution understanding” or “perceived understanding of evolution,” rather than “knowledge of evolution.” This would make clearer that the instrument captures willingness to endorse science-framed statements, not factual correctness. Such a reframing would also align more closely with the authors’ psychological interpretation that students may possess knowledge but choose whether or not to reveal it depending on context.

We agree with this and have wrestled with this throughout the course of this study. We have decided to take the recommendation to reframe it as “evolution knowledge agreement” (we played with “perceived knowledge” but perception didn’t seem quite right. It is more of an agreement with scientific statements) to better represent what the EALS is measuring.

Alternative analyses

To further strengthen the analysis, I suggest presenting the distribution of responses across Likert categories for the control versus “according to science” groups, rather than reducing them to percentages. This would more directly illustrate how framing shifts patterns of agreement. The authors could also consider statistical models that respect the ordinal nature of the data (e.g., cumulative logit models), which avoid over-interpreting the numeric distances between Likert points. Finally, scatterplots or density plots could be used to visualize how acceptance and understanding scores relate to one another in the two conditions. This would highlight the divergence without relying on potentially misleading percentage-based claims.

We considered each of these (thank you for the thorough suggestions) and decided that the stacked-bar charts best illustrated the Likert-style data in a way that was easy to interpret for the reader. We could run individual logit models on each Likert item (or goodness of fit tests for distributions), but analyzing the cumulative score seemed more appropriate.

---

## [Decision Letter · Decision Letter 2]

16 Dec 2025

We look forward to receiving your revised manuscript.

Kind regards,

Corey Cook

Academic Editor

PLOS One

Journal Requirements:

Additional Editor Comments:

Thank you for submitting your revised manuscript to PLOS ONE. Each of the reviewers feel that you have done an excellent job addressing their concerns. Reviewer 2, however, identified some very minor issues they would like to see addressed in the final version of your manuscript. Please address these final concerns and submit a revised version of the manuscript that addresses the points raised, and I will be happy to accept your manuscript for publication.

Reviewer's Responses to Questions

**Comments to the Author**

Reviewer #1: All comments have been addressed

Reviewer #2: All comments have been addressed

2. Is the manuscript technically sound, and do the data support the conclusions?

Reviewer #1: Yes

Reviewer #2: Yes

3. Has the statistical analysis been performed appropriately and rigorously?

Reviewer #1: Yes

Reviewer #2: Yes

4. Have the authors made all data underlying the findings in their manuscript fully available?

Reviewer #1: Yes

Reviewer #2: Yes

5. Is the manuscript presented in an intelligible fashion and written in standard English?

Reviewer #1: Yes

Reviewer #2: Yes

Reviewer #1: Thank you for this revision. I appreciate the way you addressed my comments, and those of the other reviewer! I think this is an important paper, well presented, ready for publication.

Reviewer #2: Thank you for the revised manuscript! I believe this work will make a meaningful contribution to the field. The re-organization of Studies 1 and 2, along with the addition of the new figure, substantially improves clarity and strengthens the overall narrative. I have only a few minor comments that I think will further enhance readability and consistency.

1. Line 121 (clarity): The sentence “In other words, two respondents with an equal knowledge of evolution may score differently on a knowledge instrument depending on their existing level of evolution acceptance” seems intended to clarify the point, but I found it difficult to understand on its own without having read the full paper, especially the Discussion

2. Table 1 and Line 292 (political ideology coding consistency): In Table 1, political ideology is presented in three categories (conservative-leaning, moderate, liberal-leaning), but the text around Line 292 appears to reference analyses based on the original 7-point scale (also described earlier as a 7-point Likert measure). I understand why you collapsed political ideology into three groups for Table 1, but I suggest clarifying this more explicitly and ensuring consistent wording across the manuscript. For example, you could state that Table 1 uses collapsed categories for descriptive purposes, while analyses use the full 7-point measure (if that is the case).

3. Include student instructions in the supplement (replicability and instructional value).

I recommend adding the exact instructions/prompts used in the study to the supplemental materials. Because the manuscript highlights the impact of adding the phrase “according to science,” including the full wording would improve transparency and make it easier for instructors and researchers to replicate or adapt the approach in future work.

4. Line 515 (add nuance about the sample/group).

The sentence would benefit from a small clarification to reflect the specific context in which this result was observed. I suggest adding wording that indicates this improvement was found in a particular study/sample (or subgroup) so the claim does not read as universally generalizable across all groups.

**Do you want your identity to be public for this peer review?** For information about this choice, including consent withdrawal, please see our Privacy Policy

Reviewer #1: No

Reviewer #2: No

---

## [Author Response · Author response to Decision Letter 3]

16 Dec 2025

Reviewer #1: Thank you for this revision. I appreciate the way you addressed my comments, and those of the other reviewer! I think this is an important paper, well presented, ready for publication.

RESPONSE: Thank you!

Reviewer #2: Thank you for the revised manuscript! I believe this work will make a meaningful contribution to the field. The re-organization of Studies 1 and 2, along with the addition of the new figure, substantially improves clarity and strengthens the overall narrative. I have only a few minor comments that I think will further enhance readability and consistency.

1. Line 121 (clarity): The sentence “In other words, two respondents with an equal knowledge of evolution may score differently on a knowledge instrument depending on their existing level of evolution acceptance” seems intended to clarify the point, but I found it difficult to understand on its own without having read the full paper, especially the Discussion

RESPONSE: Agreed. We thought it best to just delete the sentence. It does not add to the clarity. The point is stated in the sentence before.

2. Table 1 and Line 292 (political ideology coding consistency): In Table 1, political ideology is presented in three categories (conservative-leaning, moderate, liberal-leaning), but the text around Line 292 appears to reference analyses based on the original 7-point scale (also described earlier as a 7-point Likert measure). I understand why you collapsed political ideology into three groups for Table 1, but I suggest clarifying this more explicitly and ensuring consistent wording across the manuscript. For example, you could state that Table 1 uses collapsed categories for descriptive purposes, while analyses use the full 7-point measure (if that is the case).

RESPONSE: Good catch! I’m not sure why we collapsed it in the table; I suppose just for ease of interpretation. I’ve gone ahead and expanded it in the table to match all our analyses and avoid any confusion.

3. Include student instructions in the supplement (replicability and instructional value).

I recommend adding the exact instructions/prompts used in the study to the supplemental materials. Because the manuscript highlights the impact of adding the phrase “according to science,” including the full wording would improve transparency and make it easier for instructors and researchers to replicate or adapt the approach in future work.

RESPONSE: We have included the modified survey in the supplemental file 1. That is a good idea. There were no additional instructions or prompts. The survey was simply administered just as it appears in the supplement. So, we have nothing else to include.

4. Line 515 (add nuance about the sample/group).

The sentence would benefit from a small clarification to reflect the specific context in which this result was observed. I suggest adding wording that indicates this improvement was found in a particular study/sample (or subgroup) so the claim does not read as universally generalizable across all groups.

RESPONSE: Thank you. We added further clarification: “…as was done in this study using the EALS instrument with undergraduate Christian students…”

---

## [Editor Report · Decision Letter 3]

17 Dec 2025

Conflation between knowledge and acceptance may contribute to the knowledge gap between Judeo-Christian and non-religious people

PONE-D-25-06450R3

Dear Dr. Jensen,

We’re pleased to inform you that your manuscript has been judged scientifically suitable for publication and will be formally accepted for publication once it meets all outstanding technical requirements.

Kind regards,

Corey Cook

Academic Editor

PLOS One

---

## [Editor Report · Acceptance letter]

PONE-D-25-06450R3

PLOS One

Dear Dr. Jensen,

I'm pleased to inform you that your manuscript has been deemed suitable for publication in PLOS One. Congratulations! Your manuscript is now being handed over to our production team.

Kind regards,

on behalf of

Dr. Corey Cook

Academic Editor

PLOS One